# Association between well-being and compliance with COVID-19 preventive measures by healthcare professionals: A cross-sectional study

Shimoni Urvish Shah[1,2], Evelyn Xiu Ling Loo[3,4], Chun En Chua[5], Guan Sen Kew[5,6], Alla Demutska[7], Sabrina Quek[5], Scott Wong[8], Hui Xing Lau[4], En Xian Sarah Low[8], Tze Liang Loh[9], Ooi Shien Lung[10], Emily C. W. Hung[11], M. Masudur Rahman[12], Uday C. Ghoshal[13], Sunny H. Wong[14], Cynthia K. Y. Cheung[15], Ari F. Syam[16], Niandi Tan[17], Yinglian Xiao[17], Jin-Song Liu[18], Fang Lu[19], Chien-Lin Chen[20], Yeong Yeh Lee[21,22], Ruter M. Maralit[23], Yong-Sung Kim[24], Tadayuki Oshima[25], Hiroto Miwa[25], Kewin Tien Ho Siah[4,5], Junxiong Pang[1,2]*

1 Saw Swee Hock School of Public Health, National University of Singapore and National University Health System, Singapore, Singapore, 2 Centre for Infectious Disease Epidemiology and Research, National University of Singapore, Singapore, Singapore, 3 Department of Paediatrics, Yong Loo Lin School of Medicine, National University of Singapore, Singapore, Singapore, 4 Singapore Institute for Clinical Sciences (SICS), Agency for Science, Technology and Research (A*STAR), Singapore, Singapore, 5 Department of Medicine, Yong Loo Lin School of Medicine, National University of Singapore, Singapore, Singapore, 6 Division of Gastroenterology & Hepatology, Department of Medicine, National University Hospital, Singapore, Singapore, 7 Department of Clinical Psychology, James Cook University, Singapore, Singapore, 8 Department of Medicine, Ng Teng Fong General Hospital, Singapore, Singapore, 9 Department of Otorhinolaryngology, Head and Neck Surgery, Universiti Putra Malaysia, Selangor, Malaysia, 10 Department of Anaesthesiology, Columbia Asia Hospital, Miri, Sarawak, Malaysia, 11 Cambridge Paediatrics, Shatin, Hong Kong, 12 Department of Gastroenterology, Sheikh Russel National Gastroliver Institute and Hospital, Dhaka, Bangladesh, 13 Department of Gastroenterology, Sanjay Gandhi Postgraduate Institute Medical Science, Lucknow, India, 14 Department of Medicine & Therapeutics, Faculty of Medicine, The Chinese University of Hong Kong, Shatin, Hong Kong, 15 Department of Medicine, State Key Laboratory of Pharmaceutical Biotechnology, The University of Hong Kong, Hong Kong, China, 16 Division of Gastroenterology, Department of Internal Medicine, Faculty of Medicine, University of Indonesia, Jakarta, Indonesia, 17 Departments of Gastroenterology and Hepatology, The First Affiliated Hospital, Sun Yat-sen University, Guangzhou, China, 18 Department of Gastroenterology, Wuhan Union Hospital of Huazhong University of Science and Technology, Wuhan, Hubei, China, 19 Xiyuan Hospital, China Academy of Chinese Medical Sciences, Beijing, China, 20 Department of Medicine, Buddhist Tzu Chi Hospital and University School of Medicine, Hualien, Taiwan, 21 St George & Sutherland Clinical School, University of New South Wales, Sydney, Australia, 22 Gut Research Group, Faculty of Medicine, Universiti Kebangsaan Malaysia, Kuala Lumpur, Malaysia, 23 The Medical City, Metro Manila, Philippines, 24 Wonkwang Digestive Disease Research Institute, Gut and Food Healthcare, Wonkwang University School of Medicine, Iksan, South Korea, 25 Division of Gastroenterology and Hepatology, Department of Internal Medicine, Hyogo College of Medicine, Nishinomiya, Hyogo, Japan

* pangv@hotmail.com

⬛ OPEN ACCESS

**Data Availability Statement:** All relevant data are within the manuscript and its Supporting information files.

## Abstract

### Importance

Knowledge and attitude influence compliance and individuals' practices. The risk and protective factors associated with high compliance to these preventive measures are critical to enhancing pandemic preparedness.

**Funding:** JP - 1) NUS COVID-19 RESEARCH SEED FUNDING (NUSCOVID19RG-43) 2) Lloyd's Register Foundation Institute for the Public Understanding of Risk (IPUR_FY2020_RES_02_PANG) The funders had no role in study design, data collection and analysis, decision to publish, or preparation of the manuscript.

**Competing interests:** The authors have declared that no competing interests exist.

## Objective

This survey aims to assess differences in mental health, knowledge, attitudes, and practices (KAP) of preventive measures for COVID-19 amongst healthcare professionals (HCP) and non-healthcare professionals.

## Design

Multi-national cross-sectional study was carried out using electronic surveys between May-June 2020.

## Setting

Multi-national survey was distributed across 36 countries through social media, word-of-mouth, and electronic mail.

## Participants

Participants ≥21 years working in healthcare and non-healthcare related professions.

## Main outcome

Risk factors determining the difference in KAP towards personal hygiene and social distancing measures during COVID-19 amongst HCP and non-HCP.

## Results

HCP were significantly more knowledgeable on personal hygiene (AdjOR 1.45, 95% CI -1.14 to 1.83) and social distancing (AdjOR 1.31, 95% CI -1.06 to 1.61) compared to non-HCP. They were more likely to have a positive attitude towards personal hygiene and 1.5 times more willing to participate in the contact tracing app. There was high compliance towards personal hygiene and social distancing measures amongst HCP. HCP with high compliance were 1.8 times more likely to flourish and more likely to have a high sense of emotional (AdjOR 1.94, 95% CI (1.44 to 2.61), social (AdjOR 2.07, 95% CI -1.55 to 2.78), and psychological (AdjOR 2.13, 95% CI (1.59–2.85) well-being.

## Conclusion and relevance

While healthcare professionals were more knowledgeable, had more positive attitudes, their higher sense of total well-being was seen to be more critical to enhance compliance. Therefore, focusing on the well-being of the general population would help to enhance their compliance towards the preventive measures for COVID-19.

## Introduction

COVID-19 was first reported in Wuhan in December 2019. It was declared a public health emergency of international concern by the World Health Organization (WHO) on January 30, 2020 [1]. In March 2020, the COVID-19 outbreak was characterized as a pandemic to emphasize the urgency among all countries to detect, test, and build comprehensive strategies to prevent the spread of COVID-19 [2].

Prevention and public health measures are of utmost importance to reduce the spread of this disease [3] especially due to the lack of vaccine and limited treatment options at the time of the study. Some of the personal protective measures that have been implemented to prevent or minimize the spread of SARS-CoV-2 have been social distancing and good hygiene practices [4, 5].

Social distancing aims to prevent the spread of infections by reducing clustering and interactions in a community [6]. Since COVID-19 is transmitted by respiratory droplets through close contact with infectious individuals [7], social distancing is of critical importance in establishing control and has been a consistent feature of every national response to the COVID-19 pandemic. Some examples of social distancing include staying indoors, school closures, working from home where possible, and avoiding social gatherings [8].

Good hand hygiene practices can reduce the spread of respiratory diseases such as SARS-CoV, MERS-CoV, and influenza virus as they can survive on surfaces for extended periods, but it has not been proven to reduce SARS-CoV-2 transmission [9]. A systematic review on hand hygiene shows that the effectiveness of hand hygiene practices in preventing influenza and its transmission in the community is insufficient. However, due to its proven efficacy in general infectious disease prevention and control, it is still critical to adopt good hand hygiene practices as a general preventive measure [10].

The success of any preventive strategy depends on public adherence and individual willingness to take precautions which may be influenced by global factors such as news media or local factors such as infected family members or friends [11]. Many studies and surveys are being carried out by countries to understand people's attitudes and perception of COVID-19 and their association with knowledge, protective behaviors and practices [12]. However, very few studies and surveys have been conducted at a global level to understand the factors related to compliance towards various public health measures and differences in perceptions and practices between those that work in health services compared to other sectors. Guidelines, advisories, and preventive measures for diseases are issued generically to all people. However, for these to be more effective and acceptable at a community level, it is important to address the differences in perception of people in health services and other services for them to be more effective and acceptable COVID-19 has had an impact on the well-being of everyone but largely on population at risk which includes healthcare professionals [13, 14]. It is important to understand if mental health and well-being of a person affects their attitude towards complying to various preventive measures which are a key in containing the ongoing pandemic.

Questions addressing knowledge, attitude and practices (KAP) were adapted from the Health Belief Model which has been identified as a quick method to directly and quantitatively identify individual belief profiles that can help in addressing various public health preventive measures and promote education [15]. Using a combination of the Health Belief Model and the Mental Health Continuum—Short Form (MHC-SF), the survey aimed to assess the global differences in mental health and KAP of healthcare professionals (HCP) versus non-healthcare professionals (non-HCP) with respect to personal hygiene and social distancing during the COVID-19 pandemic. The survey also aimed to assess the risk factors associated with compliance towards preventive measures and the role of well-being amongst HCP.

## Methods

### Study design and data collection

This was a cross-sectional study involving 36 countries globally from May–June 2020. Participants aged 21 years and above were invited to participate in an anonymized survey through social media platforms such as Facebook ads, Instagram, WhatsApp, and word of mouth and

electronic mails. The survey was administered via the mySurvey platform (Verint Systems Inc., New York, USA) (link -https://mysurvey.nus.edu.sg/EFM/se/543BE5C2182BB4F7) and was hosted by the National University of Singapore.

## Questionnaire

The questionnaire was initially developed in the English language and then translated to other languages (including Chinese, Indonesian Bahasa, Malay, Bengali and Korean) and subsequently back-translated to resolve any discrepancies. The questionnaire has 4 main sections: 1) demographics, 2) KAP on personal hygiene, 3) KAP on social distancing, and 4) the biopsychosocial impact on participants. All questions related to KAP were adapted from the Health Belief Model and were developed by the authors of the study. The questions on attitude addressed the respondents perceived susceptibility to COVID-19 and their practices addressed their perceived response efficacy towards personal hygiene and social distancing measures. A summary of items in the questionnaire assessed are provided in S1 Table in S1 File and is briefly described below.

There were two items for the section on personal hygiene (score range 0-1/item) to assess knowledge: transmission mode of COVID-19 and the medium that could inactivate SARS-CoV-2. Likewise, there were two items to assess attitude, i.e., interest in increasing their knowledge and wearing a facemask to protect themselves and others (score range 0-1/item). To assess practices, all responses were in a 5-point Likert scale (never, seldom, 50% of the time, most of the time, always). The score for each item was totaled (sum score = 40) and averaged for this section. All 8 items were further dichotomized into low or high compliance (most of the time or always) to assess the respondent's compliance towards a specific hygiene behavior.

For the section on social distancing, there was one item on knowledge (score range 0–1) and five items on attitude (score range 0-1/item). For practices, compliance was assessed similarly to personal hygiene for 1 item while the remaining 3 items emphasized compliance to specific behaviors (including how often do they go out of the house in a week, how many people do they meet face-to-face every day and on average, the number of places they go to in a day). The score for each item was subsequently totaled (sum score = 20) and averaged for this section.

The psychological impact of the COVID-19 pandemic to HCP was assessed using the MHC-SF [16]. This is a 14-item questionnaire with three components: emotional, social, and psychological well-being. Respondents were further categorized as either flourishing or not-flourishing [17, S1 Table in S1 File]. For each item in MHC-SF, participants were asked to rate their feelings in the past month on a 6-point Likert scale (never, once or twice a month, about once a week, two or three times a week, almost every day, every day). This tool has been validated in many different languages and countries such as Italy, South Korea and South Africa [18–20]. A continuous score ranging from 0–70 was computed, and a score greater than 75% indicating a high total well-being level. Scores for emotional, social and psychological well-being were categorized as high and low.

## Data analysis

For categorical variables, frequency and percentages were recorded and for continuous variables, mean and standard deviation. Univariate analysis of the association of studied variables with HCP vs. non-HCP was assessed using the chi-square test and independent t-test. A sub-analysis for HCP was carried out to see any differences between high compliance and low compliance. Multivariate analysis was performed using a multivariate logistic regression model

with significance at p-value <0.05. All analysis was performed with IBM SPSS statistics software v26 (Chicago, IL, USA).

## Ethics

As our study was completely anonymous, it posed no more than minimal risks to respondents and waiver of informed consent would not adversely affect the rights or welfare of study subjects. It qualified for exemption from Singapore's National Health Group (NHG) Domain Specific Review Board (DSRB) ethics review (2020/00470). It was made clear to participants that by completing the questionnaire, they were giving implied consent for collected information to be used for research purpose.

## Results

### Demographic

There were a total of 2,703 respondents from 36 countries including 40.5% HCP and remaining 59.5% belonging to primarily professional (15.5%), administrative (12.8%), finance & insurance (7.4%) and engineering (7.0%) sectors. The majority of the cohort especially HCP (43.6%) were from Singapore. Table 1 shows the significant demographic differences between HCP and non-HCP.

### Knowledge, Attitude, and Practices (KAP)

**Knowledge.** *Personal hygiene*. Higher proportion of HCP knew that COVID-19 cannot be transmitted by mosquito bites (90.1% vs. 83.8%, p-value—<0.001) and SARS-CoV-2 can be inactivated by soap and alcohol disinfectant (97.8% vs. 96.3%, p-value—0.040). A significantly higher mean knowledge score was observed amongst HCP ((mean—1.88, SD– 0.35 vs. mean —1.80, SD– 0.44 (p-value—<0.001)) compared to non-HCP

*Social distancing*. The higher proportion of HCP knew that >1 or 2m was the distance to maintain socially to prevent transmission of COVID-19 (96.4% vs. 95.1%, p-value—<0.001) (S2 Table in S1 File).

**Attitude.** *Personal hygiene*. A higher proportion of HCP felt that wearing facemask was important as it protected them and others from being infected with COVID-19 (97.4% vs. 95.9%, p-value—0.027). A significantly higher positive attitude score was seen amongst HCP ((mean—1.67, SD– 0.64 vs. mean—1.55, SD– 0.74, p-value—<0.001)).

*Social distancing*. An overall positive attitude towards social distancing was seen in the cohort while significantly higher proportion of HCP was willing to participate in the contact tracing app (80.6% vs. 73.9%, p-value—<0.01). No difference in attitude towards social distancing was seen between HCP and non-HCP [S2 Table in S1 File].

**Practices.** *Personal hygiene*. An increased level of compliance towards personal hygiene practices was seen amongst HCP in washing their hands with soap or alcohol- based disinfectant >5 times/day (81.3% vs. 68.1%, p-value—<0.001), covering their mouth while sneezing or coughing (96.0% vs. 93.5%), p-value– 0.006), wearing a mask when they have flu-like symptoms even before the COVID-19 pandemic started (42.1% vs. 38.3%, p-value—0.050), avoiding touching their eyes, nose and mouth (77.8% vs. 69.6%, p-value—<0.001) and wiping surfaces and objects with disinfectant regularly (60.9% vs. 47.5%, p-value—<0.001). A significantly higher mean score for personal hygiene practices was seen amongst HCP ((mean– 33.4, SD– 4.58 vs. mean– 31.8, SD– 4.96, p-value—<0.001)).

*Social distancing*. A high level of compliance towards social distancing measure of avoiding to stand or sit close to people was observed equally amongst HCP and non-HCP. For three

**Table 1. Demographic differences between HCP and non-HCP.**

| Variable | HCP (N = 1,096) n (%) | Non-HCP (N = 1,607) n (%) | p-value |
|---|---|---|---|
| **Age (years) mean (SD)** | 37.7 (10.7) | 41.0 (13.8) | **<0.001** |
| **Gender** | | | **0.001** |
| Male | 364 (33.2) | 666 (41.4) | |
| Female | 732 (66.8) | 941 (58.6) | |
| **Race** | | | **<0.001** |
| Chinese | 580 (52.9) | 958 (59.6) | |
| Other[a] | 516 (47.1) | 649 (40.4) | |
| **Region of residence[b]** | | | **<0.001** |
| East Asia | 267 (24.4) | 495 (30.8) | |
| South East Asia (SEA) | 760 (69.3) | 1052 (65.5) | |
| Others | 69 (6.3) | 60 (3.7) | |
| **Educational Level** | | | **<0.001** |
| Secondary School (10 years) or lower | 56 (5.1) | 200 (12.4) | |
| Pre-University | 112 (10.2) | 233 (14.5) | |
| Tertiary–Undergraduate/ Postgraduate | 928 (84.7) | 1174 (73) | |
| **Current Employment** | | | **<0.001** |
| Full time | 1011 (92.2) | 988 (61.5) | |
| Part time | 61 (5.6) | 99 (6.2) | |
| Not working | 24 (2.2) | 520 (32.4) | |
| **Does your job require physical interaction with many people (Yes)** | 962 (87.8) | 824 (51.3) | **<0.001** |
| **Housing** | | | **<0.001** |
| Dormitory/Nursing | 21 (1.9) | 85 (5.3) | |
| Government Housing with 2 or more rooms | 429 (39.1) | 443 (27.6) | |
| Private apartment or condominium/landed property | 646 (58.9) | 1079 (67.1) | |
| **No. of household members** | | | 0.098 |
| < 5 | 722 (65.9) | 1098 (68.3) | |
| ≥ 5 | 374 (34.1) | 509 (31.7) | |
| **Any elderly people (65y) or young children (<12y) at home (Yes)** | 513 (46.8) | 720 (44.8) | 0.162 |
| **Any serious medical condition[c] (Yes)** | 118 (10.8) | 223 (13.9) | **<0.001** |
| **Have you been diagnosed with COVID-19? (Yes)** | | | **0.008** |
| No | 1081 (98.6) | 1560 (97.1) | |
| Pending results | 5 (0.5) | 6 (0.4) | |
| **Do you have any friend or family member who is infected by COVID-19? (Yes))** | 111 (10.1) | 134 (8.3) | 0.064 |
| What is your preferred source of obtaining information with regards to COVID 19? | | | **0.005** |
| Messaging platforms (e.g. WhatsApp/ SMS/ Telegram) from friends | 105 (9.6) | 189 (11.8) | |
| Newspaper (hardcopy) | 19 (1.7) | 42 (2.6) | |
| Online news websites/ apps | 557 (50.8) | 793 (49.3) | |
| Social media e.g. Facebook/ Instagram/ Twitter | 245 (22.4) | 288 (17.9) | |
| TV News | 105 (9.6) | 189 (11.8) | |

[a] Bengali, Caucasian, Filipino, Indian, Japanese, Korean, Malay, others

[b] East Asia: China, Hong Kong, Taiwan, South Korea, Macau, Japan; SEA: Brunei, Cambodia, Indonesia, Malaysia, Myanmar, Philippines, Singapore, Thailand, Vietnam; Others: Australia, Austria, Bangladesh, Canada, France, Georgia, India, Lebanon, Malawi, Mali, Netherlands, Reunion, Romania, Saudi Arabia, Tanzania, Turkey, United Arab Emirates, United Kingdom, United States.

[c] examples include diabetes mellitus, hypertension, hyperlipidemia, lung disease, heart disease, immunocompromised, chronic kidney disease, chronic liver disease, gastrointestinal disease, cancers

additional social distancing practices, lesser proportion of HCP had gone out of the house more than 7 times (7.7% vs. 10.5%, p-value– 0.007) excluding for work, while a greater proportion of HCP met > 20 people face-to-face (<1m apart) every day, excluding from own household (24.1% vs. 7.5%, p-value—<0.001) and went to ≥3 places in a day, excluding home (12.7% vs. 11.9%, p-value—<0.001). A significantly lower mean score for social distancing practices was seen amongst HCP (mean 14.5, SD– 2.64 vs. mean– 15.5, SD– 2.78, p-value—< 0.001) [S2 Table in S1 File].

## Mental health and well being

The proportion of HCP who thought they will never get infected with COVID-19 in the next one month was significantly lower (24.9% vs. 33.2%, p-value—<0.001)) and who were flourishing were significantly higher (74.8% vs. 68. 6%, p-value—<0.001)) as compared to non-HCP. A higher sense of total well-being was seen amongst HCP (38.2% vs. 33.7%, p-value—0.009) with higher level of emotional well-being (48.4% vs. 45.6%, p-value—0.081) and psychological well-being (45.4% vs. 42%, p-value 0.042) as well as a higher sense of social well-being (36.3% vs. 29.9%, p-value <0.001). HCP also had a higher mean score (mean- 46.2, SD-14.5 vs. mean—43.9, SD-15.0, p-value—<0.001) out of a maximum score of 70 for total well-being [S2 Table in S1 File].

## KAP risk factors associated with health and non-health related professions

After adjusting for the demographic variables that were significantly different between HCP and non-HCP, multivariate logistic regression analysis (Table 2) showed that HCP were significantly more knowledgeable on personal hygiene (AdjOR 1.45, 95% CI [1.14–1.83]) and social distancing (AdjOR 1.31, 95% CI [1.06–1.61]). HCP were 1.21 times more likely to have a positive attitude towards personal hygiene and 1.46 times more willing to participate in contact tracing app. HCP were 4.29 times more likely to have met >20 people every day outside of their household and were 2.25 times more likely to go to >4 places every day.

HCP were 1.79 times more likely to shows high compliance, 1.5 times more likely to think that their probability of getting COVID-19 in the next 1 month was >25%—< 75%. In terms of well-being, they were 1.25 times more likely to have a higher sense of total well-being and 1.33 times more likely to have a high sense of social well-being. Flourishing, emotional, and psychological well-being were not significantly different between HCP and non-HCP.

## Demographics and KAP related to HCP's compliance

**Demographic and Knowledge, Attitude, and Practices (KAP).**   HCP that showed high compliance had a significantly higher proportion of females, non-Chinese and a lower level of education than HCP that showed low compliance [Table 3].

Similar high scores were observed for knowledge and attitudes towards personal hygiene amongst HCP with different compliance levels. A similar HCP across compliance levels knew that >1 or 2m distance was ideal for maintaining effective social distancing (95.2% vs. 96.9%). The overall score for social distancing attitude was significantly higher (3.59 vs. 3.47, p-value– 0.014) for HCP with high compliance, who had a significantly higher proportion of respondents willing to participate in the contact tracing app (85.3% vs. 79.0%, p-value—0.022). Although insignificant, a higher proportion of HCP with high compliance believed that social distancing measures were important to reduce the spread of COVID-19 (99.3% vs. 97.8%, p-value– 0.189). HCP with high compliance had a higher proportion that went to no places on an average, excluding home (16.5% vs. 10.1%, p-value—0.003). A higher proportion of HCP with high compliance thought they will not get infected by COVID-19 (35.2% vs. 21.5%, p-

**Table 2. Multivariate regression analysis for difference between HCP and non-HCP.**

| Question | AdjOR (95% CI) [a] | p-value |
|---|---|---|
| **Personal Hygiene Knowledge** | | |
| COVID-19 CANNOT be transmitted by | | |
| Mosquito bites (ref) | | **0.005** |
| Door hands and hand-phone surfaces | 0.60 (0.25–1.45) | 0.258 |
| Sneezing and rubbing of eyes | 1.08 (0.68–1.73) | 0.728 |
| Not sure | **0.52 (0.35–0.75)** | **0.001** |
| Which medium can kill COVID-19? | | |
| Soap and alcohol disinfectant (ref) | | 0.210 |
| Hot water | 0.65 (0.16–2.62) | 0.545 |
| Hand dryers | 0.87 (0.34–2.23) | 0.773 |
| Not sure | **0.46 (0.22–0.97)** | **0.042** |
| **Personal Hygiene Knowledge Score** | 1.45 (1.14–1.83) | **0.003** |
| **Social Distancing Knowledge** | | |
| How far apart should people stand or sit? (ref Incorrect) | 1.31 (1.06–1.61) | **0.012** |
| **Personal Hygiene Attitude** | | |
| Wearing a facemask is important during COVID-19 Pandemic | | |
| I DO NOT think that wearing a facemask is important (ref) | | **0.048** |
| Because government ordered me to wear a facemask | 0.66 (0.21–2.13) | 0.490 |
| Because my family members asked me to wear a facemask | 0.12 (0.01–1.29) | 0.081 |
| Because we can protect our self and others from COVID-19 | 1.22 (0.46–3.25) | 0.680 |
| **Personal Hygiene Attitude Score** | 1.21 (1.06–1.39) | **0.006** |
| **Social Distancing Attitude** | | |
| Would you willingly participate in the contact tracing app? (ref No) | 1.46 (1.17–1.82) | **0.001** |
| **Personal Hygiene Practices** | | |
| How often do you wash your hands with soap or alcohol- based disinfectant a day? (ref Low compliance) | 1.82 (1.46–2.27) | **<0.001** |
| Do you cover your mouth when you sneeze or cough? (ref Low compliance) | 1.27 (0.82–1.97) | 0.279 |
| Do you usually wear a mask when you have flu-like symptoms before the COVID-19 pandemic? (ref Low compliance) | 1.22 (1.00–1.47) | **0.045** |
| Do you AVOID touching your eyes nose and mouth during COVID-19 pandemic? (ref Low compliance) | 1.45 (1.17–1.79) | **0.001** |
| Do you wipe surfaces and objects with disinfectant regularly? (ref Low compliance) | 1.54 (1.28–1.85) | **<0.001** |
| **Personal Hygiene Practice Score** | 1.07 (1.04–1.09) | **<0.001** |
| **Social Distancing Practice** | | |
| How often do you go out of the house in a week (excluding going out for work)? | | |
| Never (ref) | | 0.092 |
| 1–2 times | 1.07 (0.77–1.49) | 0.685 |
| 3–4 times | 0.95 (0.65–1.38) | 0.780 |
| 5–6 times | 1.30 (0.85–1.98) | 0.219 |
| More than 7 times | 0.75 (0.48–1.16) | 0.192 |
| How many people do you meet face-to-face (<1m) apart everyday (excluding own household)? | | |
| 0 (ref) | | **<0.001** |
| 1–5 | 1.39 (1.08–1.80) | **0.012** |
| 6–10 | 1.81 (1.31–2.51) | **<0.001** |
| 11–20 | 2.50 (1.69–3.69) | **<0.001** |
| >20 | 4.29 (3.05–6.03) | **<0.001** |

(*Continued*)

**Table 2.** (Continued)

| Question | AdjOR (95% CI) [a] | p-value |
|---|---|---|
| On average, how many places do you go in a day (excluding home)? | | |
| 0 (ref) | | **<0.001** |
| 1–2 | 2.57 (1.98–3.33) | **<0.001** |
| 3–4 | 2.19 (1.49–3.21) | **<0.001** |
| >4 | 2.25 (1.23–4.08) | **0.008** |
| **Social Distancing Practice Score** | | |
| **Compliance (ref Low)** | 1.79 (1.42–2.27) | **<0.001** |
| **Mental Health** | | |
| What do you think your probability of getting COVID19 is in the next 1 month? | | |
| 0%, I will not get infected by COVID-19 (ref) | | **0.017** |
| <25% | 1.22 (0.97–1.52) | 0.081 |
| <50% | 1.47 (1.09–1.97) | **0.010** |
| <75% | 1.95 (1.16–3.28) | **0.012** |
| 100% | 2.78 (0.70–11.02) | 0.145 |
| Effects of social distancing on mental health (ref Not flourishing) | 1.22 (0.98–1.50) | 0.066 |
| Total well-being (ref Low) | 1.25 (1.02–1.52) | **0.032** |
| Social well-being (ref Low) | 1.33 (1.09–1.64) | **0.005** |
| Psychological well-being (ref Low) | 1.08 (0.89–1.31) | 0.419 |
| **Total well-being Score** | 1.01 (1.00–1.02) | **0.002** |

[a] Adjusted for age, gender, race, region of residence, education level, employment type, housing, job requiring physical interaction with many people, suffering from serious medical condition, been diagnosed with COVID-19 and source of information.

value—<0.001), were flourishing (82.4% vs. 72.3%, p-value—<0.001) and had high sense of total well-being (53.1% vs. 33.3%, p-value—<0.001) including emotional (60.4% vs. 44.3%, p-value—<0.001), social (48.7% vs. 32.2%, p-value—<0.001) and psychological (59.3% vs. 40.8%, p-value—<0.001) [S3 Table in S1 File].

**Factors associated with high compliance amongst HCP.** After adjusting for the demographic variables that were significantly different between HCP showing high and low compliance, multivariate logistic regression analysis showed that HCP with high compliance were more likely to have gone to 0 places on an average in a day, excluding home, and were more likely to think they would not get COVID-19 in the next 1 month. HCP with high compliance were 1.86 times more likely to flourish, 2.33 times more likely to have a sense of total well-being, including emotional (AdjOR 1.94, 95% CI [1.44–2.61]), social (AdjOR 2.07, 95% CI [1.55–2.78]) and psychological (AdjOR 2.13, 95% CI [1.59–2.85]) [Table 4].

## Discussion

This study evaluated the differences in KAP, mental health status, and risk factors of compliance towards personal hygiene and social distancing among healthcare professionals (HCP) and non-HCP during the mid-COVID-19 pandemic.

HCP had a higher level of knowledge for personal hygiene and social distancing, which corroborated findings among healthcare workers in Henan, China [21]. Non-HCP were more likely to be unsure about the transmission of COVID-19 and the method to inactivate SARS-CoV-2. Experienced frontline HCW with higher education and training in COVID-19 showed

**Table 3. Demographic differences among HCP with high compliance and low compliance.**

| Variable | Low compliance (N = 823) n (%) | High compliance (N = 273) n (%) | p-value |
|---|---|---|---|
| **Age (years) mean (SD)** | 37.4 (10.7) | 38.8 (10.8) | 0.069 |
| **Gender** | | | **<0.001** |
| Male | 301 (36.6) | 63 (23.1) | |
| Female | 522 (63.4) | 210 (76.9) | |
| **Race** | | | **0.003** |
| Chinese | 457 (55.5) | 123 (45.1) | |
| Other[a] | 366 (44.5) | 150 (54.9) | |
| **Region of residence[b]** | | | 0.278 |
| East Asia | 193 (23.5) | 74 (27.1) | |
| SEA | 574 (69.7) | 186 (68.1) | |
| Others | 56 (6.8) | 13 (4.8) | |
| **Educational Level** | | | **<0.001** |
| No formal education | 1 (0.1) | 1 (0.4) | |
| Secondary School (10 years) | 33 (4.0) | 21 (7.7) | |
| Pre-University | 72 (8.7) | 40 (14.7) | |
| Tertiary–Undergraduate/ Postgraduate | 717 (87.1) | 211 (77.3) | |
| **Current Employment** | | | 0.334 |
| Full time | 755 (91.7) | 256 (93.8) | |
| Part time | 47 (5.7) | 14 (5.1) | |
| Not working | 21 (2.6) | 3 (1.1) | |
| **Does your job require physical interaction with many people (Yes)** | 717 (87.1) | 245 (89.7) | 0.287 |
| **Housing** | | | 0.156 |
| Dormitory/Nursing | 17 (2.1) | 4 (1.5) | |
| Government Housing with 2 or more rooms | 309 (37.5) | 120 (44.0) | |
| Private apartment or condominium/landed property | 497 (60.4) | 149 (54.6) | |
| **No. of HH members** | | | 0.418 |
| < 5 | 548 (66.6) | 174 (63.7) | |
| ≥ 5 | 275 (33.4) | 99 (36.3) | |
| **Any elderly people (65y) or young children (<12y) at home (Yes)** | 387 (47.0) | 126 (46.2) | 0.834 |
| **Any serious medical condition[c] (Yes)** | 93 (11.4) | 25 (9.2) | 0.573 |
| **Have you been diagnosed with COVID-19? (Yes)** | | | 0.531 |
| Yes | 9 (1.1) | 1 (0.4) | |
| Pending results | 4 (0.5) | 1 (0.4) | |
| **Do you have any friend or family member who is infected by COVID-19? (Yes)** | 88 (10.7) | 23 (8.5) | 0.550 |
| **What is your preferred source of obtaining information with regards to COVID 19?** | | | 0.243 |
| Messaging platforms (e.g. WhatsApp/ SMS/ Telegram) from friends | 86 (10.4) | 19 (7.0) | |
| Newspaper (hardcopy) | 13 (1.6) | 6 (2.2) | |
| Online news websites/ apps | 420 (51.0) | 137 (50.2) | |
| Social media e.g. Facebook/ Instagram/ Twitter | 185 (22.5) | 60 (22.0) | |
| TV News | 1119 (4.5) | 51 (18.7) | |

[a] Bengali, Caucasian, Filipino, Indian, Japanese, Korean, Malay, others

[b] East Asia: China, Hong Kong, Taiwan, South Korea, Japan; SEA: Brunei, Cambodia, Indonesia, Malaysia, Myanmar, Singapore, Vietnam; Others: Australia, Bangladesh, Canada, France, India, Malawi, Reunion, Romania, Saudi Arabia, Tanzania, United Kingdom, United States.

[c] examples include diabetes mellitus, hypertension, hyperlipidemia, lung disease, heart disease, immunocompromised, chronic kidney disease, chronic liver disease, gastrointestinal disease, cancers

**Table 4. Multivariate regression analysis for factors associated with compliance amongst HCP.**

| Question | AdjOR (95% CI) [a] | p-value |
|---|---|---|
| **Social Distancing Attitude** | | |
| Would you willingly participate in the contact tracing app? (ref No) | 1.41 (0.96–2.07) | 0.081 |
| **Social Distancing Attitude Score** | 1.23 (0.99–1.51) | 0.056 |
| **Social Distance Practice** | | |
| **On average, how many places do you go in a day (excluding home)?** | | |
| 0 (ref) | | **0.019** |
| 1–2 | 0.61 (0.40–0.91) | **0.015** |
| 3–4 | 0.44 (0.23–0.83) | **0.012** |
| >4 | 1.09 (0.46–2.62) | 0.834 |
| **Mental Health** | | |
| What do you think your probability of getting COVID19 is in the next 1 month? | | |
| 0%, I will not get infected by COVID-19 (ref) | | **0.005** |
| <25% | 0.59 (0.42–0.82) | **0.002** |
| <50% | 0.47 (0.30–0.73) | **0.001** |
| <75% | 0.75 (0.39–1.39) | 0.361 |
| 100% | 0 (0) | 0.999 |
| Effects of social distancing on mental health—Flourishing (ref Not flourishing) | 1.86 (1.30–2.67) | **0.001** |
| Total well-being (ref Low) | 2.33 (1.74–3.12) | **<0.001** |
| Emotional well-being (ref Low) | 1.94 (1.44–2.61) | **<0.001** |
| Social well-being (ref Low) | 2.07 (1.55–2.78) | **<0.001** |
| Psychological well-being (ref Low) | 2.13 (1.59–2.85) | **<0.001** |
| **Total well-being Score** | 1.03 (1.02–1.04) | **<0.001** |

[a] Adjusted for gender, race and education level.

a better KAP on perceived risk levels, indicating that increased awareness and education needs to be imparted to the general community in order for them to understand the risk factors of COVID-19 [22]. Knowledge is essential to establish the importance of prevention, promote positive behavior and attitude, affecting the effectiveness of coping strategies and behaviors to a certain extent [23].

An overall higher level of positive attitude towards personal hygiene was seen amongst HCP compared to non-HCP. These attitudes include wearing of facemask and willingness to participate in contact tracing app, both important measures to prevent the spread of COVID-19 [24]. By participating in the contact tracing app, which has been implemented in few countries, one is able to track cases and their contacts to swiftly quarantine potential cases and prevent further spread of the disease. We need to understand the reason for hesitation to take part in the contact tracing app amongst the general community and ensure that mask wearing is not only important during a pandemic but even under general conditions when one is unwell. Non-HCP in general, showed less compliance towards personal hygiene practices, thus advocating for stricter rules and more efforts looking at behavioral changes amongst the general population.

The attitude towards social distancing was similar amongst HCP and non-HCP, likely because of the fear that their family and friends may get infected with COVID-19. Similar findings were observed in a North American and European study which found that protecting others, self and community were the most common motivations in engaging in social distancing [25]. Amongst the general population, washing hands and keeping away from crowded places were seen as 'the right thing to do' and the main motivation to comply [26].

Frequent hand washing and avoidance of shaking hands were the dominant practices seen in the general population in Southwest Ethiopia [27]. Whilst our study had similar findings, we also found higher level of compliance amongst HCP towards personal hygiene practices like washing hands, covering mouth while sneezing, wearing mask while displaying flu-like symptoms, avoiding touching their eyes, nose and mouth and during wiping surfaces and objects with disinfectant regularly. A review by Mathur P emphasizes the importance of hand hygiene in reducing the risk of cross-transmission of infections [28] which makes it even more important for the non-HCP to comply. While all respondents avoided standing or sitting close to people, HCP was more sociable and meeting more people, and going to more places in a day, most likely due to their work load and professional demands, making self-isolation difficult. Although HCP had a higher proportion of wearing a mask when they had flu-like symptoms even before the COVID-19 pandemic, the proportions were still very low (42.1% vs. 38.3%). Due to limitations of data collection, we were unable to identify the different types of HCP which may help us understand if certain practices, attitudes were more prevalent amongst certain section of healthcare professionals. Importance of wearing a facemask when sick needs to be stressed globally irrespective of the current pandemic situation as there is evidence that population-wide use of face masks can delay pandemics and reduce the reproduction number, thereby helping to contain an outbreak [29]. This practice along with several other preventive behaviors can be achieved by messages focusing on "protecting your community" as concluded by Capraro and Barcelo [30].

Our study saw multiple sources being used to obtain COVID-19 information similar to a survey being carried out among HCP in the United Kingdom [31]. Sources used were significantly different; electronic news and social media being more prevalent amongst HCP and messaging platforms and TV news being more common amongst non-HCP. This information helps target relevant public health messages through a suitable platform for the right cohort of people. As one study concluded that messages with a positive language were likely to be adhered to by people and that people with leadership roles should be engaged in motivating their colleagues and informal social circles by sharing public health messages [32].

A study in Italy [33] revealed that the healthcare workers perceive having 2.5 times higher risk of COVID-19 infection than the general population, similar to 1.9 times seen in our study. While flourishing mental health, emotional and psychological well-being were similar across cohorts, a higher sense of overall well-being and social well-being was seen amongst HCP. However, this is contradictory to higher levels of depression and stress seen amongst HCWs that are assisting COVID-19 patients [34]. A study among nurses showed a negative correlation between perceived stress and happiness scores [13] which can help explain the increased social well-being seen amongst HCP in our study as their professional role may provide them with a large sense of satisfaction and meaning towards protecting the community from COVID-19 even though occupational stress level is likely higher. Job satisfaction was also observed to be significantly associated with a high level of total positive mental health status, and so was the workplace environment [35]. A review on impact of COVID-19 on mental health showed student status, unemployment, presence of chronic illness, poor self-rated health were some of the risk factors that predicted stress in the general community [14]. Efforts need to be directed towards the mental health of the community, especially in times of lockdown and social distancing where support from friends and families can be minimal, aggravating loneliness and producing negative long-term health consequences that affect ones social and mental well-being [36].

This is the first study, to our best knowledge, that focused on risk factors among HCP with high/low compliance behaviors. HCPs who are females and those with a pre-university level of education are more likely to have high compliance behavior. This was noted in other studies

where male staff tend to have a higher incidence of unsatisfactory hand washing than females [37] and physicians were more likely than nurses and allied health professionals to need external reminders for hand hygiene [38]. This highlights the importance of targeting public health interventions among those at-risk populations to strengthen high compliance behavior further. In addition, HCP with high compliance behavior were more likely to flourish and have a high sense of emotional, social, and psychological well-being. This was similar to findings from Hong Kong COVID-19 health information survey that found lower stress levels and less anxiety and depressive symptoms to be positively associated with perceived compliance towards social distancing measures in the general population [39]. Another study amongst college students concluded that compliance towards social distancing measures was not predicted by risk tolerance or increased risk factors of being infected [40]. Therefore, further research is still required to verify the causal risk factors associated with high compliance behavior amongst the healthcare professionals and general population to help successfully implement preventive measures.

## Limitations

Our study has some limitations. Firstly, over-simplification of findings or social desirability bias due to closed responses or responding as per what may seem correct can lead to poor reliability and validity of the findings. Secondly, questions based on their past practices could have led to recall bias. Although data was gathered from 36 countries, findings may not be truly representative of the demographics of each country making the findings less generalizable. Another limitation was the lack of assessment on the validity and reliability of the survey instrument in each of the country involved, which could have provided a more accurate interpretation of the findings and a more robust instrument. Being a cross-sectional study, a causal relationship of risk factors with compliance cannot be established. Lastly, overestimation of the risk effect is observed as we have not captured the risk factors contributing to KAP and mental health of the study sample.

## Conclusions

Healthcare professionals were more knowledgeable, showed increased motivation towards practicing personal hygiene and social distancing and had better total well-being compared to non-healthcare professionals. A high level of total well-being may attribute to the high compliance behavior amongst healthcare professionals. Based on the results we believe that by focusing on the total well-being of the general population we can help in increasing their compliance towards various preventive measures.

## Supporting information

**S1 File. Tables of survey distributed to the respondents, KAP and mental health differences between HCP and non-HCP and between HCP with low and high compliance.** (DOCX)

## Author Contributions

**Conceptualization:** Kewin Tien Ho Siah, Junxiong Pang.

**Data curation:** Shimoni Urvish Shah, Chun En Chua, Guan Sen Kew, Alla Demutska, Sabrina Quek, Scott Wong, Hui Xing Lau, En Xian Sarah Low, Tze Liang Loh, Ooi Shien Lung, Emily C. W. Hung, M. Masudur Rahman, Uday C. Ghoshal, Sunny H. Wong, Cynthia K.

Y. Cheung, Ari F. Syam, Niandi Tan, Yinglian Xiao, Jin-Song Liu, Fang Lu, Chien-Lin Chen, Yeong Yeh Lee, Ruter M. Maralit, Yong-Sung Kim, Tadayuki Oshima, Hiroto Miwa, Kewin Tien Ho Siah.

**Formal analysis:** Shimoni Urvish Shah.

**Funding acquisition:** Junxiong Pang.

**Methodology:** Shimoni Urvish Shah, Junxiong Pang.

**Project administration:** Evelyn Xiu Ling Loo, Kewin Tien Ho Siah, Junxiong Pang.

**Resources:** Evelyn Xiu Ling Loo, Kewin Tien Ho Siah, Junxiong Pang.

**Supervision:** Evelyn Xiu Ling Loo, Kewin Tien Ho Siah, Junxiong Pang.

**Validation:** Shimoni Urvish Shah, Evelyn Xiu Ling Loo, Junxiong Pang.

**Writing – original draft:** Shimoni Urvish Shah.

**Writing – review & editing:** Evelyn Xiu Ling Loo, Chun En Chua, Guan Sen Kew, Alla Demutska, Sabrina Quek, Scott Wong, Hui Xing Lau, En Xian Sarah Low, Tze Liang Loh, Ooi Shien Lung, Emily C. W. Hung, M. Masudur Rahman, Uday C. Ghoshal, Sunny H. Wong, Cynthia K. Y. Cheung, Ari F. Syam, Niandi Tan, Yinglian Xiao, Jin-Song Liu, Fang Lu, Chien-Lin Chen, Yeong Yeh Lee, Ruter M. Maralit, Yong-Sung Kim, Tadayuki Oshima, Hiroto Miwa, Kewin Tien Ho Siah, Junxiong Pang.

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
