## [Decision Letter · Decision Letter 0]

29 Apr 2021

PONE-D-21-09372

Association between Well-Being and Compliance with COVID-19 Preventive Measures by Healthcare Professionals: A cross-sectional study

PLOS ONE

Dear Dr. Pang,

Thank you for submitting your manuscript to PLOS ONE. After careful consideration, we feel that it has merit but does not fully meet PLOS ONE’s publication criteria as it currently stands. Therefore, we invite you to submit a revised version of the manuscript that addresses the points raised during the review process.

Please find below the reviewer's comments, as well as those of mine.

We look forward to receiving your revised manuscript.

Kind regards,

Valerio Capraro

Academic Editor

PLOS ONE

Additional Editor Comments:

I have now collected one review from one expert in the field. I was unable to find a second reviewer. However, I am familiar with the topic of this manuscript, therefore I feel confident in making a decision with only one review. The reviewer thinks that the paper can potentially make a valuable contribution, but suggests a major revision. After reading the review and the article, I agree with the reviewer, and therefore I would like to invite you to revise your work for Plos One. Besides the reviewer's comments, that should all be addressed, I would like to add two more comments from my own reading: (i) regarding gender differences in compliance to public health recommendations, another paper making this point is: Capraro and Barcelo, 2020; (ii) In the general discussion, you might find useful the perspective article on what social and behavioral science can do to support pandemic response, published by Van Bavel et al. in Nature Human Behaviour. Of course, it is not a requirement to cite these papers, I am mentioning them because they look very related to your work and therefore you might find them useful.

I am looking forward for the revision.

References

Capraro, V., & Barcelo, H. (2020). The effect of messaging and gender on intentions to wear a face covering to slow down COVID-19 transmission. Journal of Behavioral Economics for Policy, 4, Special Issue 2, 45-55.

Van Bavel, J. J., et al. (2020). Using social and behavioural science to support COVID-19 pandemic response. Nature Human Behaviour, 4, 460-471.

Journal Requirements:

2. Please include additional information regarding the survey or questionnaire used in the study and ensure that you have provided sufficient details that others could replicate the analyses. For instance, if you developed a questionnaire as part of this study and it is not under a copyright more restrictive than CC-BY, please include a copy, in both the original language and English, as Supporting Information.  If the original language is written in non-Latin characters, for example Amharic, Chinese, or Korean, please use a file format that ensures these characters are visible.

3. Thank you for including your ethics statement:  "No individually identifiable information was obtained during the questionnaire and confidentiality, and anonymity was maintained. Ethics approval was obtained for this study. The NHG DSRB reference number for this study is 2020/00470.".   

a.) Please amend your current ethics statement to include the full name of the ethics committee/institutional review board(s) that approved your specific study.

b.) Please provide additional details regarding participant consent. In the ethics statement in the Methods and online submission information, please ensure that you have specified (1) whether consent was informed and (2) what type you obtained (for instance, written or verbal, and if verbal, how it was documented and witnessed). If your study included minors, state whether you obtained consent from parents or guardians. If the need for consent was waived by the ethics committee, please include this information.

Reviewers' comments:

Reviewer's Responses to Questions

**Comments to the Author**

1. Is the manuscript technically sound, and do the data support the conclusions?

Reviewer #1: Yes

2. Has the statistical analysis been performed appropriately and rigorously? 

Reviewer #1: Yes

3. Have the authors made all data underlying the findings in their manuscript fully available?

Reviewer #1: Yes

4. Is the manuscript presented in an intelligible fashion and written in standard English?

Reviewer #1: Yes

5. Review Comments to the Author

Reviewer #1: This is an interesting and relevant study exploring factors associated with transmission-preventative behaviour among different occupational groups. It is a useful contribution, however I have a number of comments to consider:

Introduction:

- You could add more about the practical implications of this research on HCPs i.e. why is studying the behaviours of this occupational group so important? Maybe related to maintaining standards of health services? protecting the most vulnerable? role models?

- Related to the above, perhaps clarify if the emphasis is on exploring who and what behaviours need to be targeted to hinder the spread of COVID-19? And/or if the emphasis is on understanding what predicts transmission-preventative behaviour in HCP vs. non-HCP? (this would inform the narrative of the discussion)

- Not a lot is said about the wellbeing focus of the survey in the introduction - why is this important to explore?

- Was any particular theory used to inform the focus on KAP?

Method:

- Were the KAP items adapted from an existing questionnaire or developed by the authors?

- Perhaps comment on the methodological robustness of the scales e.g. reliability/validity

Results:

- Did you collect data on the types of HCP within this group? Primary/secondary sector; domiciliary care?

Discussion:

- Since the discussion focuses on the implications of the research in terms of supporting transmission-preventative behaviour among HCP and non-HCP, maybe the title and aims should be tweaked to reflect this?

- Line 323-324 "However, this needs further research and understanding" - perhaps expand on this, I was not sure what this related to?

- Line 340-349: could explore more as to why non-HCP might experience lower sense of wellbeing (furloughed? home-working?)

- Limitation also of using cross-sectional design

- The conclusion could build more upon the 'take home' message of the study, in an applied sense i.e. a key practical implication

6. PLOS authors have the option to publish the peer review history of their article (what does this mean?). If published, this will include your full peer review and any attached files.

Reviewer #1: **Yes: **Emma Berry

---

## [Author Response · Author response to Decision Letter 0]

10 May 2021

PONE-D-21-09372

Association between Well-Being and Compliance with COVID-19 Preventive Measures by Healthcare Professionals: A cross-sectional study

PLOS ONE

Dear Valerio and Emma,

Thank you for your comments and feedback. We have revised and edited the manuscript accordingly and have attached the following documents in the system for review:

• 'Response to Reviewers' which responds to each point raised

• A marked-up copy of the manuscript labeled 'Revised Manuscript with Track Changes'

• The revised paper without tracked changes titled 'Manuscript'

• The supporting information document - supplementary material

Do let us know if you have any further comments or feedback. Thank you once again for considering our manuscript for publication in PLOS One. 

Best regards,

Junxiong Pang

---

## [Decision Letter · Decision Letter 1]

18 May 2021

PONE-D-21-09372R1

Association between Well-Being and Compliance with COVID-19 Preventive Measures by Healthcare Professionals: A cross-sectional study

PLOS ONE

Dear Dr. Pang,

Thank you for submitting your manuscript to PLOS ONE. After careful consideration, we feel that it has merit but does not fully meet PLOS ONE’s publication criteria as it currently stands. Therefore, we invite you to submit a revised version of the manuscript that addresses the points raised during the review process.

We look forward to receiving your revised manuscript.

Kind regards,

Valerio Capraro

Academic Editor

PLOS ONE

Journal Requirements:

Additional Editor Comments (if provided):

The reviewer is happy with the revision, but suggests one final improvement before publication. Please address this comment at your earliest convenience.

I am looking forward to receiving the final version.

Reviewers' comments:

Reviewer's Responses to Questions

**Comments to the Author**

1. If the authors have adequately addressed your comments raised in a previous round of review and you feel that this manuscript is now acceptable for publication, you may indicate that here to bypass the “Comments to the Author” section, enter your conflict of interest statement in the “Confidential to Editor” section, and submit your "Accept" recommendation.

Reviewer #1: (No Response)

2. Is the manuscript technically sound, and do the data support the conclusions?

Reviewer #1: Yes

3. Has the statistical analysis been performed appropriately and rigorously? 

Reviewer #1: Yes

4. Have the authors made all data underlying the findings in their manuscript fully available?

Reviewer #1: Yes

5. Is the manuscript presented in an intelligible fashion and written in standard English?

Reviewer #1: Yes

6. Review Comments to the Author

Reviewer #1: Thank you for addressing my comments/suggestions. Just one further minor suggestion to add a reference for the Health Belief Model and a little more detail around the items being adapted from this model and the rationale for this (even though the aim, as you say, was not to replicate/test the model).

7. PLOS authors have the option to publish the peer review history of their article (what does this mean?). If published, this will include your full peer review and any attached files.

Reviewer #1: **Yes: **Emma Berry

---

## [Author Response · Author response to Decision Letter 1]

19 May 2021

PONE-D-21-09372

Association between Well-Being and Compliance with COVID-19 Preventive Measures by Healthcare Professionals: A cross-sectional study

PLOS ONE

Dear Valerio,

Thank you for your feedback and considering our manuscript for submission to PLOS ONE. We have incorporated and addressed the comment raised by Emma. We have also added the following reference to the existing reference list:

15. Costa MF. Health belief model for coronavirus infection risk determinants. Rev Saude Publica. 2020;54:47

The following files have been submitted in the system:

• A rebuttal letter that responds to each point raised labeled 'Response to Reviewers'.

• A marked-up copy of the manuscript labeled 'Revised Manuscript with Track Changes'

• An unmarked version of the revised paper without tracked changes labeled 'Manuscript'

Do let us know if you require any further clarifications. We would like to thank you once again for this opportunity. 

Best regards,

Dr. Junxiong Pang

---

## [Editor Report · Decision Letter 2]

24 May 2021

Association between Well-Being and Compliance with COVID-19 Preventive Measures by Healthcare Professionals: A cross-sectional study

PONE-D-21-09372R2

Dear Dr. Pang,

We’re pleased to inform you that your manuscript has been judged scientifically suitable for publication and will be formally accepted for publication once it meets all outstanding technical requirements.

Kind regards,

Valerio Capraro

Academic Editor

PLOS ONE
---

## [Editor Report · Acceptance letter]

27 May 2021

PONE-D-21-09372R2 

Association between well-being and compliance with COVID-19 preventive measures by healthcare professionals: A cross-sectional study 

Dear Dr. Pang:

I'm pleased to inform you that your manuscript has been deemed suitable for publication in PLOS ONE. Congratulations! Your manuscript is now with our production department. 

Kind regards, 

on behalf of

Dr. Valerio Capraro 

Academic Editor

PLOS ONE